# Different Protocols for Low Whole-Body Vibration Frequency for Spasticity and Physical Performance in Children with Spastic Cerebral Palsy

**DOI:** 10.3390/children10030458

**Published:** 2023-02-26

**Authors:** Punnee Peungsuwan, Uraiwan Chatchawan, Wanida Donpunha, Pisamai Malila, Thanyaluck Sriboonreung

**Affiliations:** 1School of Physical Therapy, Faculty of Associated Medical Sciences, Khon Kaen University, Khon Kaen 40002, Thailand; 2Research Center in Back, Neck, Other Joint Pain and Human Performance (BNOJPH), Faculty of Associated Medical Sciences, Khon Kaen University, Khon Kaen 40002, Thailand; 3Physical Therapy Department, Faculty of Associated Medical Sciences, Chiang Mai University, Chiang Mai 50000, Thailand

**Keywords:** whole-body vibration, frequency protocol, spasticity, cerebral palsy, strength, balance

## Abstract

Background: Whole-body vibration (WBV) is a therapeutic exercise tool that can be used in children with cerebral palsy (CP). A low vibration frequency with different protocols has been suggested, but no optimal dose has been explicitly indicated. We aimed to determine the superiority of a gradually increased 7–18 Hz WBV protocol over a static 11 Hz WBV and the immediate and short-term effects of WBV training on improving spasticity, functional strength, balance, and walking ability in children with spastic CP. Methods: Twenty-four participants with CP (mean age: 11.5 ± 2.9 years) were randomly allocated into protocols of a static 11 Hz vibration frequency group (SVF) or one that increased from a 7 to an 18 Hz vibration frequency (IVF) (*n* = 12/group). The WBV programmes were completed for 30 min/session/day to identify immediate effects, and the short-term programme then continued for four days/week for eight weeks. Results: Modified Ashworth Scale scores significantly and immediately improved in the IVF group (hip adductor and knee extensor, *p* < 0.05), and after eight weeks showed significant improvement in the SVF group (ankle plantar flexor, *p* < 0.05). Within groups, the Five Times Sit to Stand Test (FTSTS), the Time Up and Go Test and the Functional Reach Test significantly improved in the SVF group, whereas only the FTSTS improved in the IVF group (*p* < 0.05). There were no significant between-group differences at the eight-week postintervention, except reduced spasticity. Conclusions: A protocol of 7–18 Hz WBV seems to offer superior immediate results in terms of improved spasticity; however, a static 11 Hz protocol appears to offer superior results after eight weeks, although the two protocols did not differ significantly in effects on physical performance. This finding may facilitate preparations to normalise muscle tone before functional mobility therapy. The study results may support future studies about the dose-response of WBV frequency.

## 1. Introduction

Cerebral palsy (CP) is described as a group of permanent disorders of the development of movement and posture that cause activity limitations [1]. A type of spastic diplegia is the most common form of CP, which is found in 30–40% of spastic CP patients [2]. This type of CP usually affects muscle control and coordination, and affected people have increased muscle tone, an imbalance of muscle tone, and muscle weakness, including the instability of the trunk as one of the most important problems. The main treatment goal is to reduce muscle tone and stiffness in the joints and increase range of motion, strength, and balance. There are various regular therapy methods—such as prolonged passive stretching, casting, exercise, hydrotherapy, and physical therapy—that lead to improved muscle strength, motor ability, and functional performance.

Recently, whole-body vibration (WBV) has been used as an alternative therapeutic exercise in a wide variety of rehabilitations, such as stroke [3], Parkinson’s disease [4], older adults [5], and CP [6]. Thus, WBV has become more popular in rehabilitation settings. Previous studies reported the positive effects of a WBV intervention in patients with CP regarding decreased spasticity, improved range of movement, muscle strength, balance, walking ability, and function performance [7,8,9,10,11]. Models of an acute effect and a short-term effect of vibrational training have shown improvements in these outcomes. A systematic review of WBV concluded that it was a promising intervention, but there was a paucity of evidence, particularly for young children with CP [12]. The WBV loading parameters by the mode of vibration transmitted to the body (frequency, amplitude, peak-to-peak displacement, acceleration) and by the subjects’ characteristics influenced the effects of the fitness parameters [13].

Previous research has focused on the role of various parameters of whole-body vibration training for children with spastic cerebral palsy—such as adjusting the vibration frequency and varying the time per session and duration—and other parameters, such as amplitude and wave direction, are usually fixed. In such WBV training, the frequencies were 25–40 Hz (6 min/s, 8 weeks) [6], 12–18 Hz (9 min/s, 3 months) [9], 12–18 Hz (9 min/s, 12 weeks) [10], 20 Hz (6 min/s, 6 weeks) [11], 5–25 Hz (18 min/s, 8 weeks) [13], and 13–18 Hz (12 min/s, 6 months) [14]. Previous studies found improvements in spasticity [6,10,11], muscle strength [10,11], gait ability [10,13,14], and GMFM88 [10], but no change in walking balance [10] or bone [14].

Usage doses of WBV vary widely, and it is impossible to suggest an optimal intensity for achieving therapeutic intervention for people with CP [15]. It has been hypothesized that low-amplitude and low-frequency mechanical stimulation of the human body is a safe and effective way to improve muscle strength [15]. Such wave frequency has been suggested as an important factor in determining the intensity of WBV applications [15]. The frequency protocol of WBV training is an important factor in planning a programme for patients with CP. The WBV intervention protocol differed among studies that used a frequency dose that also varied in individuals with CP. The frequency ranged generally from 5 to 40 Hz. The duration of vibrations in each session largely varied among studies, from 3–20 min. The total sessions and the total weeks also varied generally, from a single session to three–five sessions/week, and weeks range from 2–12. Several previous studies have used an under 20 Hz frequency, which reveals the main issue of how the use of frequency protocols depends on each subject’s characteristics.

To date, a low-frequency protocol appears to be the most effective WBV training regimen; however, different protocols may be linked to different clinical outcomes for children with spastic CP, and these differences are still unknown. It is therefore of interest to determine whether two different protocols of static and gradually increased low frequency during the training programme, by maintaining equal total training volume, are more effective in improving physical performance in children with spastic CP. The aim of this study was therefore to determine whether a gradually increased 7–18 Hz WBV protocol is superior to a static 11 Hz WBV and identify the immediate single session and short-term effects of WBV training on improvement in spasticity, functional strength, balance, and walking ability in children with spastic CP. Our hypothesis was that a higher 7–18 Hz frequency of WBV would be superior to a lower 11 Hz frequency in both the immediate and short-term effects of WBV training on improving spasticity, functional strength, balance, and walking ability in children with spastic CP.

## 2. Materials and Methods

### 2.1. Study Design

The study was a parallel randomised trial with two experimental groups designed as the low vibration frequency of a static 11 Hz (SVF) group and an incremental from 7 to 18 Hz (IVF) group. A low range of vibration frequencies for children with CP has been suggested [14]. This study examined the immediate single effect and 8-week effect of WBV training on changes in leg spasticity, functional strength, and balance. This study was approved by the Khon Kaen University Ethics Committee for Human Research, Thailand (HE592246). All participants and their parents signed a written informed consent form before the study began.

### 2.2. Participants

Twenty-four children with spastic CP (range of age: 7–14 years) were recruited from a special school in Khon Kaen Province, Thailand. The participant flow diagram (Figure 1) shows the random allocation into the SVF and IVF groups (*n* = 12 per group). The sample size calculation was based on a previous study [11]. Sampling randomisation was stratified according to the Gross Motor Function Classification System (GMFCS) levels I to III and the age groups of 7–10 years and 11–14 years, which created strata for each age group and GMFCS level (2 × 3 strata). The total sample size, *n* = 24, was allocated to each group of the strata proportionally to their sizes, and a researcher, not an assessor, made the strata to fit the study purposes. The inclusion criteria were as follows: the type of spastic diplegia, Modified Ashworth Scale (MAS) scores of bilateral lower extremities equal to or greater than one, ability to walk for at least 10 m with or without walking aids, ability to comprehend informed consent protocols and to take instructions from observers during the study and no participation in any exercise programme within 3 months before or during the study. The subjects received routine physiotherapy once a week. Exclusion criteria were participants who regularly exercised more than 2 times/week or 150 min/week, had received botulinum toxin injections to the lower limbs within the past 6 months, used medicine to reduce spasticity, or had other complications, such as pain, arthritis, neuromuscular and musculoskeletal disorders, and cardiovascular and pulmonary diseases. The baseline outcomes were measured before all participants were allocated by block randomisation into each group using a closed envelope.

### 2.3. Protocols

Several previous WBV studies with different frequencies have been used for people with CP, starting at 5 Hz and increasing to 30 Hz; most studies have remained under 20 Hz [7,8,9,10,11]. Low-frequency vibrations (<20 Hz) have been used to impart relaxation and to change resting length [16]. Before the experiment, the researchers tested to determine different forms of frequency for safety and to reach the target dose for the study participants. If participants complained of muscle fatigue or joint pain during increasing intensity, that frequency was terminated and was a decision-making condition. The vibration frequencies were therefore selected to obtain two protocols for individuals: a static 11 Hz form (comfortable sense zone) and an increased form of 7 Hz to 18 Hz (light to heavy sense zone). Applying static lower-body WBV training, each participant stood with equal weight borne on both feet and held the handles on a WBV platform (AIKO vibrator, ETF-001CG, Singapore, Thailand) with a vertical displacement of 2–4 mm. For the WBV programmes shown in Table 1, in the SVF group, the frequency started at 5 Hz and progressed by 0.5–1 Hz to the targeted 11 Hz within 2 min, and this level was continuously maintained until a 5-min set was completed and 6 sets were completed. In the IVF group, the frequency was started at 7 Hz and gradually increased 2–5 Hz until it reached the targeted 18 Hz for a total of 3 min/set and a total of 10 sets. All participants performed a total of 30 min/day WBV training and rested for 1 min while seated on the chair between sets. The WBV frequency was administered depending on the patient’s capacity to tolerate the vibration. Both protocols were managed by two assistant therapists. Additionally, the participants were instructed to receive weekly physiotherapy programmes, which consisted of passive muscle stretching and independent walking training with and without aids for 30 min/day.

### 2.4. Outcome Measures

The modified Ashworth scale (MAS) was used to measure the participants’ leg spasticity. Functional muscle strength was represented by the Five Times Sit to Stand Test (FTSTS). The Functional Reach Test (FRT) and the Timed Up and Go Test (TUG) indicate static and dynamic physical balance, respectively, and the 10 Metre Walk Test (10 MWT) represents walking speed. The MAS was only measured as an immediate single-bout effect and repeated after 8 weeks of WBV training. Functional ability outcomes were measured at baseline and after 8 weeks of WBV training. Two blinded assessors evaluated the MAS and functional strength and balance. Only MAS measured the immediate effect of a WBV session after completing 30 min. Afterwards, the WBV training was continuously conducted 4 days/week for 8 weeks in both groups. All outcomes were measured on the seventh day after the completed 8-week WBV training to affirm the sustained results within a week.

#### 2.4.1. Modified Ashworth Scale

MAS is the primary outcome of this study. The MAS scores are defined as follows: 0 indicates no increase in muscle tone; (1) is a slight increase in muscle tone, manifested by a catch and release or by minimal resistance at the end of the range of motion when the affected part is moved in flexion or extension; (1+) is a slight increase in muscle tone, manifested by a catch, followed by minimal resistance throughout the remainder (less than half) of the ROM; (2) is a more marked increase in muscle tone throughout the remainder of the ROM, but the affected part is easily moved; (3) is a considerable increase in muscle tone passivity, with movement being difficult; and (4) is an affected part rigid in flexion or extension [16]. Both sides of the hip adductor, quadricep, hamstring, and soleus muscles were evaluated. Before the outcome measures, the assessor tested the intra-rater reliability (intraclass correlation coefficients, ICC = 0.96). Spasticity is usually disturbed by changes in position; therefore, after the last WBV set was finished, participants were carefully transferred by wheelchair from the vibrator platform to a supine position before the MAS tested the immediate WBV effect.

#### 2.4.2. Five Times Sit to Stand Test

The FTSTS test is a reliable tool for measuring the functional strength of the lower limbs and balance ability [17]. Participants started by sitting on a chair with their arms crossed over their chests, and they were asked to stand up as quickly as possible five times without using arm support. The test began with the word “Go” and stopped when they sat with their buttocks placed on a chair after the fifth performance. The time was recorded. The test used the lowest time recorded from the two trials.

#### 2.4.3. Functional Reach Test

The FRT is defined as the maximum distance that a participant can reach forward while maintaining a fixed base of support in the standing position. This test indicates functional static balance. Reach was recorded, measuring from the initial point to the end point, while the subject’s arm was extended parallel to a yardstick attached to the wall. The ICC for this test, 0.94, showed high intra-rater reliability [18]. Three trials were performed with a seated rest of 3–5 s between test intervals, and the averaged data among the three trials were recorded.

#### 2.4.4. Timed Up and Go Test

The TUG test is an outcome measure used to assess functional dynamic balance. The reliability of the TUG test in children with CP was found to have a high ICC of 0.99 for within-session reliability and 0.99 for test–retest reliability [19]. Participants were seated with their feet flat on the floor in such a way that their hips and knees remained at 90° flexion. In the measurement, the participant was asked to stand up from a chair, walk 3 m, turn around a marked cone, walk back to the chair, and sit down again. During the test, subjects were reminded to walk not to run, as “this is not a race”. Time was monitored from when the subject stood up until the subject’s bottom touched the seat. The test was conducted 3 times with 1 min rest periods between each test, and the 3 values were averaged for analysis. Participants were allowed to use walking aids, such as canes or walkers, and wear their regular footwear or orthosis devices.

#### 2.4.5. Ten Metre Walk Test

The 10 MWT was conducted on a 14-metre walkway with 2 m marked at the start and end points of the walkway to allow for acceleration and deceleration. Before the test, an examiner requested the participant, “Please walk towards the end, at your usual speed”. When the participant stood ready at the start point, the examiner instructed “ready and go”, and the subjects walked towards the end of the walkway. The examiner started a stopwatch when the participant’s first foot crossed the plane of the 2-metre line and stopped the stopwatch when the participant’s first foot crossed the plane of the 12-metre line. During the test, one examiner walked beside the participant and encouraged him/her to continue walking. The time to reach the middle of the 10-metre walk was recorded to calculate walking speed. The test demonstrated high reliability, with an intraclass correlation of 0.81 for test–retest reliability in children with CP [20].

### 2.5. Statistical Analysis

A software package (G-Power 3.0) was used to calculate the sample size based on a primary Modified Ashworth Scale (MAS) outcome using the determination of the mean difference of a 1.0 score, and a standard deviation (0.87) which was referred to in a pilot study in our previous study [11]. A power of 80% and a significance of 5% were set. According to these criteria, 12 patients per group, or 24 patients, were recruited. All statistical analyses were performed using STATA 13.0 (StataCorp LP, College Station, TX, USA). The Shapiro–Wilk test was used to test for normal distribution among all measured variables, and parametric statistical analysis was selected for this study. The MAS is a categorical variable that is divided into 5 scales (0, 1, 1+, 2, 3, and 4) to represent types of specific meanings. The MAS data were analysed using a score for the improvement of each muscle, that is, a decreased post-test score compared with the baseline or pre-test scores at least a score of one ordinal. Improving MAS scores counted the participant numbers and showed the percentage of improvement in comparisons between groups using the two-sample test of proportion. The functional ability outcomes (FTSTS, TUG, FRT, and 10 MW) were analysed for differences within groups using a paired *t*-test and for differences between groups using the analysis of covariance (ANCOVA) and the baseline of each variable as a covariate. The null hypothesis was set as a two-tailed test, with the level of significance set at *p* < 0.05; a Bonferroni correction was used to compare different groups.

## 3. Results

### 3.1. Demographics of Participants

Twenty-four participants completed the WBV programmes, and none dropped out of the study. The participant baseline characteristics are shown in Table 2. There were no significant differences in average age, weight, height, or body mass index (BMI) between the groups. Most of the participants who used walkers were classified as GMFCS level III (66.66%).

### 3.2. Immediate and after 8-Week WBV Training Outcomes

Table 3 and Figure 2 show a comparison of the improved MAS immediately after one bout (Figure 2A) and after eight weeks of WBV training (Figure 2B) between the SVF and IVF groups. The MAS of the hip adductors of a stronger spastic side significantly decreased greater than *n* = 5, 41.67% (CI: 6.70 to 76.63%; *p* = 0.035) in seven subjects in the IVF group versus two subjects in the SVF group, and the knee extensors of a weaker spastic side significantly improved greater than *n* = 5, 41.67% (CI: 9.43 to 73.99%; *p* = 0.025) in six subjects in the IVF group versus one subject in the SVF group. In contrast, after the eight-week WBV training, the MAS of the SVF group significantly improved by 33.33% (CI: 6.66 to 60.00%, *p* = 0.029) in the ankle plantar flexor of the weaker side leg spasticity in the SVF group in four subjects versus zero, with no subject exhibiting improved spasticity in the IVF group.

The effects of the eight-week WBV training on functional strength, balance, and walking speed are shown in Table 4. Comparing within groups showed the mean difference in the FTSTS, TUG, and FRT significantly improved by −5.74 ± 3.44 s (*p* < 0.001), −2.44 ± 1.76 s (*p* < 0.001) and 3.56 ± 5.16 cm (*p* = 0.036) in the SVF group, respectively. Only the FTSTS test significantly improved in the IVF group (mean difference: −4.20 ± 3.88 cm, *p* < 0.01), and the 10 MWT revealed no significant difference. Furthermore, no significant difference was found between the groups.

## 4. Discussion

This study showed, within significant differences, that a short-term effect of the static 11 Hz frequency protocol improved FTSS, TUG, and FRT, and the 7 to 18 Hz frequency protocols improved only FTSS. No significant between-group differences, except for the immediate effect of the 7 to 18 Hz frequency protocol, decreased spasticity in hip adductors and knee extensors, while short-term WBV training showed an improved spasticity in ankle plantar flexor with the 11 Hz frequency protocol. Although the static low 11 Hz protocol showed better results, it seems that both protocols showed no difference in improvement in functional strength and balance and did not have any effect on walking ability in short-term WBV training. The results show a difference in the low-WBV protocols with patterns of static and progressively increased frequencies, with both differing in immediate effects and after short-term training. It is thus possible to suggest that a very low 11 Hz vibration frequency could improve spasticity, functional strength, and balance with eight weeks of WBV training.

After a single session of WBV training, there was a greater immediate improvement in spasticity in the large muscles, such as the hip adductors and knee extensors, in the 7 to 18 Hz group, but after eight weeks, it influenced smaller muscles, such as the ankle plantar flexor, in the 11 Hz group. However, it is worth mentioning that this is only for a proportion of children with CP. The results vary between muscle groups, but rarely did more than half of the participants respond. Similarly, a case report found improved spasticity in the ankle plantar flexor after eight-week horizontal WBV training at 1–9 Hz in children with spastic diplegia [21]. A single 20-min session of 20 Hz WBV improved the spasticity of ankle plantar flexors for 1–2 h in children with CP [22]. A previous study demonstrated that an acute modulation of motor control in spastic CP children after a 1-min bout of WBV (16–25 Hz) reduced stretch reflex activity in soleus muscle responses concomitant with increased voluntary muscular activation, improved intermuscular coordination of antagonists and increased knee joint mobility, which might be interpreted as counteracting spasticity-associated deficits in children with CP [23]. It seems that the frequency exposure on the reduced spasticity might be linked to an appropriate lower-to-upper range from under 25 Hz frequencies in children with CP. However, a previous study reported that, in adults with spastic diplegia, WBV training with 25–40 Hz was more effective in decreasing spasticity of the knee extensors than resistance training after an eight-week intervention [7].

The authors of this study evaluated only MAS test scores and immediate effect differences between groups, unlike a previous study that measured muscle strength [24]. In a previous study, the 12, 18, and 26 Hz WBV frequencies immediately and significantly improved the TUG and 10MWT in the 12 Hz and 18 Hz frequencies in children with CP [25]. We assume that using WBV in a single session with lower 20 Hz frequencies may contribute to preparing muscle tone before other rehabilitation techniques. In the short term, WBV training could improve the physical performance of children with spastic diplegia. Two protocols, 30 min/session/day of low vibration frequency and four days/week of training (120 min/week), were used in this study, considering each individual patient’s threshold, safety, and the possible effectiveness for children with spastic diplegia who were not familiar with WBV devices. Moreover, the participants had been irregularly receiving physical therapy at a special boarding school. Some children with spastic CP have hypersensitivity to vibration waves, which diminishes their ability to reach a target frequency; this was found in the present study. Therefore, a WBV programme should provide a relatively low oscillation option. Patients with disabilities may use general commercial WBV tools, which commonly emit a low-frequency vibration wave (<20 Hz).

A few studies have reported adding short-term WBV training to conventional physical therapy intervention, with frequencies ranging from 12–18 Hz [14] and 5–25 Hz [13] improving walking speed and frequencies of 12–18 Hz increasing knee extensor peak torque and stability [9] in children with CP when compared with only conventional physical therapy for an eight-week period. The present study did not find an improvement in walking speed (10MWT) after eight weeks of WBV training within either protocol group. The potential explanation depends on whether there is no additional physical therapy intervention or a higher level of GMFCS. This factor included eight subjects/group of GMFCS level III in the present study. In addition, a short WBV intervention at a low frequency had little effect on muscle activity in children with mild CP at stages 1 and 2 of the GMFCS [21]. However, a previous study reported that low-intensity frequency interventions were unlikely to improve muscle strength and functionality in children with CP [12].

There were some potential limitations to this study. We are concerned with the following methodological limitations: The MAS is an ordinal scale, and the category data was thus analysed as a proportional statistic, which is unlike previous studies. A small sample size would decrease the statistical power. The results ultimately show no statistically significant intergroup differences by a Bonferroni, which would be justified for primary and secondary outcomes. In addition, it is possible that both protocols had similar effects. The sample comprised only heterogeneous individuals with spastic diplegic CP, including a wide range of GMFCS, from levels I to III (with most patients at 66.66%, level III, use of a walker). Thompson et al.’s [26] study supports the greatest difference in muscle strength between diplegic CP independent walkers and those dependent on walking aids. In addition, both groups received a similar WBV intervention without routine physical therapy, with the only difference occurring in WBV frequency. In addition, this study cannot be compared to previous studies because of the absence of a control group (neither sham WBV treatment nor conventional physical therapy). However, it is difficult to sub-group all subjects and include a control group with such a small sample size of spastic CP children, and doing so could influence the study’s effect size. Similarly, the MAS outcome was broken down into sub-group analyses to compare stronger and weaker spastic muscles. This causes the probability of detecting the effects of fluctuating power and type I or II errors; in other words, it is the probability of rejecting or accepting the null hypothesis when it is in fact false. Two points to note: The MAS measure’s psychometric characteristics had poor inter-tester reliability, and the examiners were known at each time point in the intervention, implying a potential intervention bias. Furthermore, the study was double-blind and was set up for the participants and for a statistical data analyser in a manner intended to reduce bias and confounding factors.

There were no adverse effects on the study participants during the vibration therapy sessions or at the end of the eight weeks, except that a few participants complained of itchiness on their thighs during a frequency peak and at the end of sessions when the set was near a peak of 7 to 18 Hz. Most participants reflected that they enjoyed standing on the vibration platform because they perceived relaxation in their legs. This may keep them engaged in WBV therapy and prevent them from dropping out of the programme.

The findings mention that the 7–18 Hz WBV protocol immediately reduced spasticity; this result may be helpful to physical therapists conducting interventions before movement therapy, whereas, over the long term, therapists may consider using the static 11 Hz to improve the strength and balance performance of children with spastic diplegia. A systematic review with meta-analysis suggested that WBV may improve gait speed and standing function in children with CP and could be considered for inclusion in rehabilitation programmes [12]. WBV should also be considered as an alternative method in addition to conventional physical therapy in children with CP, especially those with moderate mental and physical impairments who are unable to walk without a walker or crutches. In other words, WBV is an available approach for patients with CP that involves a small setting, requires little time for therapy, and can support self-care at home. Thus, a model of WBV treatment may be useful for clinicians, physical therapists, and other rehabilitation specialists.

## 5. Conclusions

Immediate improvement in spasticity seems to be superior with a protocol of 7–18 Hz WBV, but after eight weeks, a static 11 Hz protocol, as well as two low-frequency WBV protocols, were associated with an improvement in functional strength in children with spastic CP. WBV protocols are also suitable, comfortable and require little effort; they may be a potential auxiliary treatment tool for these individuals. To a greater degree of interpretation, the study results may support future studies about the dose-response of WBV frequency.

## Figures and Tables

**Figure 1 children-10-00458-f001:**
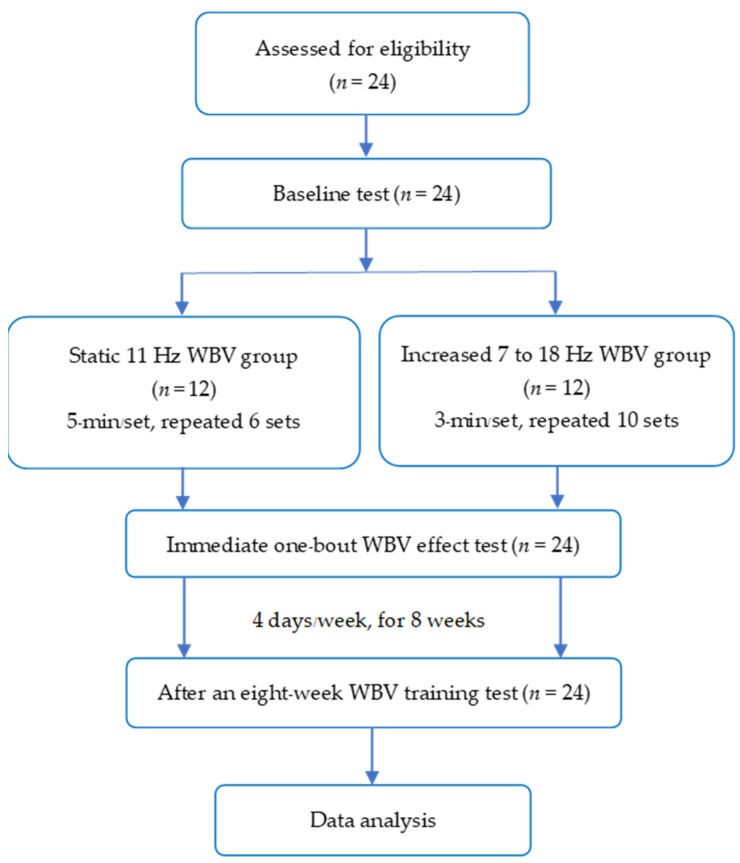
The participant’s flow chart. SVF, static vibration frequency; IVF, increased vibration frequency; WBV, whole-body vibration.

**Figure 2 children-10-00458-f002:**
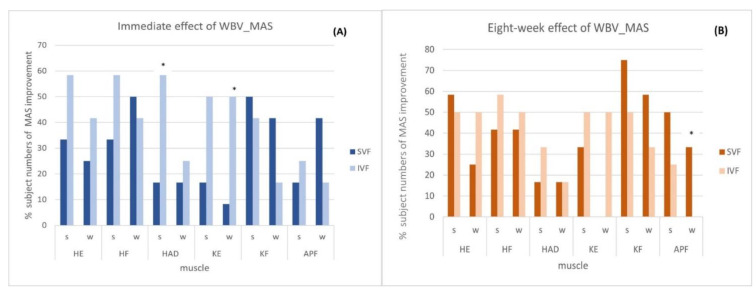
(**A**,**B**). Comparison of the MAS improvement (% of subject numbers) immediately after (**A**) and after 8 weeks (**B**) of WBV training between the SVF (*n* = 12) and IVF (*n* = 12) groups. S, stronger spastic leg side; W, weaker spastic leg side. HF, hip flexor; HE, hip extensor; HAD, hip adductor; KF, knee flexor; KE, knee extensor; APF, ankle plantar flexor. *: *p* < 0.05.

**Table 1 children-10-00458-t001:** Whole body vibration training programmes.

	SVF Group Static 11 Hz	min	IVF Group Increased 7–18 Hz	min
Step 1:	Warm-up: 5 to 7 Hz	1	7 to 10 Hz	1
Step 2:	Warm-up: 7 to <9 Hz	1	11 to 15 Hz	1
Step 3:	11 Hz	3	16 to 18 Hz	1
rest	Seated on chair	1	Seated on chair	1
Total 6 repetitions		30	Total 10 repetitions	30

**Table 2 children-10-00458-t002:** Participant characteristics.

Characteristics	SVF Group (*n* = 12)	IVF Group (*n* = 12)	*p*-Value	95%CI
Age, years	11.67 ± 2.43	11.5 ± 2.64	0.49	−3.32 to 1.65
Weight, kg	33.89 ± 11.67	38.12 ± 12.59	0.40	−14.50 to 6.06
High, cm	132.6 ± 11.10	141.9 ± 12.57	0.07	−19.35 to 0.73
BMI, kg/m^2^	19.15	18.90	0.12	−9.35 to 2.56
GMFCS I/II/III, n	2/2/8	2/2/8	-	-
No aids/walker/crutches, n	3/9/0	4/7/1	-	-

SVF group, the static 11 Hz group. IVF group, the increased 7 Hz to 18 Hz group.

**Table 3 children-10-00458-t003:** The number of decreased MAS subjects (%) and comparisons of an improved difference between groups immediately after and after 8-weeks WBV training (*n* = 12/group).

Muscles/Groups	Immediate Effects			After 8-Weeks		
	Decreased MAS *n* (%)	Improved Difference between Groups *n* (%), (95% CI)	*p*-Value	Decreased MAS *n* (%)	Improved Difference between Groups *n* (%), (95% CI)	*p*-Value
**HE**						
Stronger spastic leg						
SVF	4 (33.33)	3 (25.00)	0.219	7 (58.33)	1 (8.33)	0.682
IVF	7 (58.33)	(−13.59 to 63.59)		6 (50.00)	(−31.4 to 48.06)	
Weaker spastic leg						
SVF	3 (25)	2 (16.67)	0.387	3 (25)	3 (25)	0.206
IVF	5 (41.67)	(−20.46 to 53.79)		6 (50.00)	(−21.42 to 62.42)	
**HF**						
Stronger spastic leg						
SVF	4 (33.33)	3 (25.00)	0.219	5 (41.67)	2 (16.67)	0.414
IVF	7 (58.33)	(−13.59 to 63.59)		7 (58.33)	(−22.78 to 56.11)	
Weaker spastic leg						
SVF	6 (50.00)	1 (8.33)	0.682	5 (41.67)	1 (8.33)	0.682
IVF	5 (41.67)	(−31.4 to 48.06)		6 (50.00)	(−31.4 to 48.06)	
**HAD**						
Stronger spastic leg	2 (16.67)	5 (41.67)	0.035 *	2 (16.67)	2 (16.67)	0.346
SVF	7 (58.33)	(6.70 to 76.63)		4 (33.33)	(−17.33 to 50.67)	
IVF						
Weaker spastic leg						
SVF	2 (16.67)	1 (8.33)	0.615	2 (16.67)	0 (0)	1.00
IVF	3 (25)	(−23.99 to 40.66)		2 (16.67)	(−29.82 to 29.82)	
**KE**						
Stronger spastic leg						
SVF	2 (16.67)	4 (33.33)	0.083	4 (33.33)	2 (16.67)	0.408
IVF	6 (50.00)	(−1.95 to 68.62)		6 (50.00)	(−22.21 to 55.55)	
Weaker spastic leg SVF IVF						
	1 (8.33)	5 (41.67)	0.025 *	4 (33.33)	2 (16.67)	0.408
	6 (50.00)	(9.43 to 73.99)		6 (50.00)	(−22.21 to 55.55)	
**KF**						
Stronger spastic leg	6 (50.00)	1 (8.33)	0.682	9 (75.00)	3 (25.00)	0.206
SVF	5 (41.67)	(−31.4 to 48.06)		6 (50.00)	(−12.42 to 62.42)	
IVF						
Weaker spastic leg						
SVF	5 (41.67)	3 (25.00)	0.178	7 (58.33)	3 (25.00)	0.219
IVF	2 (16.67)	(−9.67 to 59.97)		4 (33.33)	(−13.59 to 63.59)	
**APF**						
Stronger spastic leg	2 (16.67)	1 (8.33)	0.615	6 (50.00)	3 (25)	0.206
SVF	3 (25)	(−23.99 to 40.66)		3 (25)	(−12.42 to 62.42)	
IVF						
Weaker spastic leg						
SVF	5 (41.67)	3 (25)	0.178	4 (33.33)	4 (33.33)	0.029 *
IVF	2 (16.67)	(−9.97 to 59.97)		0 (0)	(6.66 to 60.00)	

Data is presented as the number of subjects (n) and percent (%). HF, hip flexor; HE, hip extensor; HAD, hip adductor; KF, knee flexor; KE, knee extensor; APF, ankle plantar flexor. * *p* < 0.05.

**Table 4 children-10-00458-t004:** Comparison of the tests of functional strength, balance, and walking speed within and between the SVF and IVF groups.

Functional Ability Test	Groups	Within the SVF and IVF Groups’ Baselines and after 8 Weeks				Between SVF and IVF Groups after 8 Weeks ^$^	
		Baseline	After 8 Weeks	Change (95% CI)	*p*-Value	Mean Difference (95% CI)	*p*-Value
FTSTS	SVF	17.07 ± 3.66	11.33 ± 3.21	−5.74 ± 3.44 (−7.92 to −3.55)	0.0001 ^#^	−1.65 (−3.81 to 0.52)	0.13
	IVF	17.26 ± 6.00	13.06 ± 3.53	−4.20 ± 3.88 (−6.66 to −1.74)	0.003 **		
TUG	SVF	19.59 ± 10.97	17.15 ± 10.49	−2.44 ± 1.76 (−3.56 to −1.32)	0.0006 ^#^	−0.65 (−5.01 to 3.71)	0.76
	IVF	19.43 ± 10.39	17.66 ± 11.49	−1.77 ± 6.99 (−6.22 to 2.67)	0.40		
FRT	SVF	17.08 ± 08	20.64 ± 5.84	3.56 ± 5.16 (0.27 to 6.84)	0.036 *	0.53 (−3.43 to 4.49)	0.78
	IVF	17.55 ± 3.1	20.26 ± 3.35	2.71 ± 5.08 (−0.51 to 5.94)	0.09		
10 MW	SVF	13.28 ± 4.13	12.27 ± 4.61	−1.01 (−2.31 to 0.30)	0.12	−1.71 (−6.02 to 2.60)	0.42
	IVF	14.66 ± 11.55	14.81 ± 9.04	0.15 (−5.11 to 5.42)	0.95		

Data presented as mean ± standard deviation. FTSS, Five Times Sit to Stand; TUG, Time Up and Go; FRT, Functional Reach; 10 MWT, 10-min walk. *, ** and ^#^ are *p*-values of <0.05, <0.01 and <0.001, respectively. ^$^ denoted using the analysis of covariance (ANCOVA).

## Data Availability

The data is unavailable due to privacy or ethical restrictions.

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
