# Peer review of "Different Protocols for Low Whole-Body Vibration Frequency for Spasticity and Physical Performance in Children with Spastic Cerebral Palsy"

_children, 2023, doi:10.3390/children10030458_

Round 1
Reviewer 1 Report
The purpose of the study is clearly named. The measured parameters are well defined. Considering the number of participants the authors refer to an own previous study where they calculated 12 as the necessary number of participants. This refers especially to the modified Ashworth scale. In the present study there are more parameters measured beside the MAS therefore the number of participants should be explained. It would be interesting to know in which range of the MAS the improvement could be seen. It is not noted which significance the results have considering participation of the children.
It is missing which kind of physiotherapy the children had besides the whole body vibration
The last statement of the conclusion is not justified and does appear as a consequence of the study.
Line 40: it is more an imbalance of muscle tone and not only an increase muscle tone, the instability of the trunk is one of the most important problems, one of the effects of WBV is to increase strength of muscle
Author Response
Dear Reviewer 1,
Response to Reviewer 1 Comments
Point 1: Considering the number of participants the authors refer to an own previous study where they calculated 12 as the necessary number of participants. This refers especially to the modified Ashworth scale.
In the present study there are more parameters measured beside the MAS therefore the number of participants should be explained. It would be interesting to know in which range of the MAS the improvement could be seen. It is not noted which significance the results have considering participation of the children.
Response 1: Added a sentence of “The MAS is a primary outcome using sample size calculation.” to part of statistical analysis (line 230-231).
Included “Table 3” shows MAS scores as the number of the decreased MAS subjects (%) and comparisons of an improved difference between groups immediately after and after 8-weeks WBV training” (line 266-270)
Point 2: It is missing which kind of physiotherapy the children had besides the whole body vibration
Response 2: Added a sentence of “which consisted of passive muscle stretching and independent walking training with and without aids for 30 minutes/day.” (line 154-155)
Point 3: The last statement of the conclusion is not justified and does appear as a consequence of the study.
Response 3: Revised in part of the conclusion (line 385-392)
Point 4: Line 40: it is more an imbalance of muscle tone and not only an increase muscle tone, the instability of the trunk is one of the most important problems, one of the effects of WBV is to increase strength of muscle.
Response 4: Revised a sentence (line 46-47)
Regards,

Reviewer 2 Report
This is a parallel randomized trial with two experimental groups reporting on the immediate and short-term effects of whole-body vibration training with two different frequency protocols on spasticity, functional strength, balance, and walking ability in children with spastic diplegic cerebral palsy. I have some major/minor concerns that the authors need to address before re-submission. Please, find them below.
1. Abstract:- the conclusion is a repetition of the results. The final statement of the abstract should concisely summarize your study's conclusions, implications, or applications to practice and, if appropriate, can be followed by a statement about the need for additional research revealed from the findings. This should be considered to allow readers quickly grasp the major ideas and lets them know whether reading the entire paper is worthwhile.
2. Introduction:- authors may need to create some context and background and explain the role of different parameters of the whole-body vibration training for children with cerebral palsy in light of the previous studies.
3. Introduction:- The authors pinpointed the gap in the literature and made it clear what drove them to conduct the study. They indicated that “no study has compared the effectiveness of different low-frequency protocols for WBV training programs to determine the most effective exercise training regimen”. It is unclear, though, just how crucial it is to address this gap. If there have been differences in the clinical outcomes linked to various training protocols, the authors may need to underline this.
4. The inclusion criteria are not clear enough. Specify the characteristics required for study entry, such as the severity of the lesion or specific clinical/pathophysiological characteristics to help readers identify the population in which it is expected that the effect of the intervention can be shown.
5. I suggest providing a further description of the stratified randomization procedure. What was the size of each stratum? Was the allocation to each group proportional to the size of each stratum? Who performed the randomization?
6. I am concerned about the lack of control subjects (no WBV group). This is an acritical issue that affects the internal validity of the study.
7. Is this study powered enough for the study design and measured outcomes? I am not sure. You included 24 subjects, which is a considerably small sample. The critical problem is that the authors may have conducted inappropriate power analyses (although not clarified in the manuscript) which resulted in the study being grossly underpowered for dealing with inflation of the experiment-wise error rate involved in the large family of comparisons. Even a modest adjustment in alpha to handle that problem would render statistically nonsignificant almost of the comparisons presently reported as significant even though many involve effect sizes that are moderate or large. It is clearly something that would have been much better handled a priori by conducting an appropriate power analysis to balance off family-wise Type I and Type II error risk. I'm intrigued by the authors' defense.
8. How was the power analysis was conducted for these analyses? This has not been mentioned. A more properly powered investigation would have resulted in a solid conclusion, and the authors wouldn’t be left trying to explain away why such a poor balance has been struck between Type I and Type II error risk while also trying to deal with the problems in an acceptable way on a post hoc basis—something that, incidentally, does not occur presently.
9. Analysis:- you employed ANCOVA to calculate the between-group differences. Which variables have been factored as covariates is not clear?
10. Authors should include in-depth discussion focusing on the interpretation of their results, stressing their relevance to practice, and offering their viewpoint on how intended professionals (clinicians, physical therapists, and other rehabilitation specialists) make use of these data.
Author Response
Dear Reviewer 2,
Response to Reviewer 2 Comments
Point 1: Abstract:- the conclusion is a repetition of the results. The final statement of the abstract should concisely summarize your study's conclusions, implications, or applications to practice and, if appropriate, can be followed by a statement about the need for additional research revealed from the findings. This should be considered to allow readers quickly grasp the major ideas and lets them know whether reading the entire paper is worthwhile.
Response 1: Revised in part of the conclusion in the abstract (line 29-36).
Point 2: Introduction:- authors may need to create some context and background and explain the role of different parameters of the whole-body vibration training for children with cerebral palsy in light of the previous studies.
Response 2: Added the previous study reviews in part of the introduction (line 63-71).
Point 3: Introduction:- The authors pinpointed the gap in the literature and made it clear what drove them to conduct the study. They indicated that “no study has compared the effectiveness of different low-frequency protocols for WBV training programs to determine the most effective exercise training regimen”. It is unclear, though, just how crucial it is to address this gap. If there have been differences in the clinical outcomes linked to various training protocols, the authors may need to underline this.
Response 3: Revised the main gap issue in part of the introduction (line 85-87).
Point 4: The inclusion criteria are not clear enough. Specify the characteristics required for study entry, such as the severity of the lesion or specific clinical/pathophysiological characteristics to help readers identify the population in which it is expected that the effect of the intervention can be shown.
Response 4: Added the inclusion criteria in part of the 2.2 participants (line 117-119).
Point 5: I suggest providing a further description of the stratified randomization procedure. What was the size of each stratum? Was the allocation to each group proportional to the size of each stratum? Who performed the randomization?
Response 5: Added all in part of the 2.2 Participants (line 112-115).
Point 6: I am concerned about the lack of control subjects (no WBV group). This is an acritical issue that affects the internal validity of the study.
Response 6: This is a limitation of this study. An explanation of this point was added. (line 354-355)
Point 7: Is this study powered enough for the study design and measured outcomes? I am not sure. You included 24 subjects, which is a considerably small sample. The critical problem is that the authors may have conducted inappropriate power analyses (although not clarified in the manuscript) which resulted in the study being grossly underpowered for dealing with inflation of the experiment-wise error rate involved in the large family of comparisons. Even a modest adjustment in alpha to handle that problem would render statistically nonsignificant almost of the comparisons presently reported as significant even though many involve effect sizes that are moderate or large. It is clearly something that would have been much better handled a priori by conducting an appropriate power analysis to balance off family-wise Type I and Type II error risk. I'm intrigued by the authors' defense.
Response 7: Added “power level of 80 %” in part of the statistical analysis (line 229-242), and explained the reasons that may had Type I and Type II error risk in the limitation part of the discussion (line 354-364).
Point 8: How was the power analysis was conducted for these analyses? This has not been mentioned. A more properly powered investigation would have resulted in a solid conclusion, and the authors wouldn’t be left trying to explain away why such a poor balance has been struck between Type I and Type II error risk while also trying to deal with the problems in an acceptable way on a post hoc basis—something that, incidentally, does not occur presently.
Response 8: Added “power level of 80 % and an effect size of 20 %.” in part of the statistical analysis (line 229), and explanation in the reasons that may had Type I and Type II error risk in the discussion (354-364).
Point 9: Analysis:- you employed ANCOVA to calculate the between-group differences. Which variables have been factored as covariates is not clear?
Response 9: Revised in part of the statistical analysis (line 241)
Point 10: Authors should include in-depth discussion focusing on the interpretation of their results, stressing their relevance to practice, and offering their viewpoint on how intended professionals (clinicians, physical therapists, and other rehabilitation specialists) make use of these data.
Response 10: Revised in part of the discussion (line 271-282)

Round 2
Reviewer 2 Report
The authors addressed the issues and suggestions mentioned during the initial review round, which resulted in a considerable improvement in the manuscript's clarity and general presentation.
Author Response
Dear Reviewer 2,
My manuscript was thoroughly re-read and approved by a native English speaker.
Thank you so much.
